# Obstacles to inclusion and threats to civil rights: An integrative review of the social experiences of service dog partners in the United States

Sarah C. Leighton [1,2]*, Molly E. Hofer[2,3], Cara A. Miller[4], Matthias R. Mehl[2], Tammi D. Walker [2,5], Evan L. MacLean[1], Marguerite E. O'Haire [1]

1 College of Veterinary Medicine, University of Arizona, Oro Valley, Arizona, United States of America, 2 College of Science, University of Arizona, Tucson, Arizona, United States of America, 3 College of Social & Behavioral Sciences, University of Arizona, Tucson, Arizona, United States of America, 4 School of Human Services and Sciences, Gallaudet University, Washington, District of Columbia, United States of America, 5 James E. Rogers College of Law, University of Arizona, Tucson, Arizona, United States of America

* sarahleighton@arizona.edu

## Abstract

Service dogs, trained to assist people with disabilities, are known to impact their human partners' social experiences. While service dogs can act as a "social bridge," facilitating greater social connection under certain circumstances, many service dog partners also encounter challenges in social settings because of the presence of their service dog – despite legal protections. Among the most common challenges reported are experiences of stigma, discrimination, and access or service denials. This preregistered integrative review sought to synthesize empirical, theoretical, and legal literature to understand better the social experiences reported by service dog partners in the United States, including (1) civil rights experiences; (2) experiences of stigma and discrimination; and (3) broader social experiences. Following database searches and article screening, a total of $N = 43$ articles met the eligibility criteria for inclusion. Analyses were conducted in two stages: first, synthesizing quantitative and qualitative findings to explore the magnitude of social experiences reported by empirical articles and second, narrative synthesis to integrate findings across all article types. Analyses identified three themes: *Adverse Social Experiences*, *Contributing Factors*, and *Proposed Solutions*. Overall, we found consistent reports of stigma, discrimination, and access denials for service dog handlers. Additionally, these adverse experiences may be more common for service dog partners with disabilities not externally visible (i.e., invisible disabilities such as diabetes or substantially limiting mental health conditions). This integrative review highlights a pattern of social marginalization and stigmatization for some service dog partners, exacerbated by inadequate legal protection and widespread service dog fraud. These findings have implications for the individual well-being of people with disabilities partnered with service dogs and highlight a need for collective efforts to increase inclusion and access. Effective solutions likely require a multi-component approach operating at various socio-ecological levels.

**Data availability statement:** This was a pre-registered review (Open Science Framework: https://osf.io/hj9nb).

**Funding:** The research reported in this publication was funded by the University of Arizona One Health Research Initiative (SL). The content is solely the responsibility of the authors and does not necessarily represent the official views of the funders.

**Competing interests:** The authors have declared that no competing interests exist.

## Introduction

In 1990, a historic piece of civil rights legislation was enacted to combat discrimination against individuals with disabilities in the United States. The Americans with Disabilities Act (ADA) was created to safeguard the civil rights of people with disabilities by establishing "clear, consistent, and enforceable standards" [1]. The ADA was born out of the disability advocacy movement, which aims to counter the ableism and historical marginalization that people with disabilities have long faced [2]. It was enacted after decades of activism and advocacy by individuals with disabilities who fought to raise public awareness of the barriers they faced; these barriers included inaccessible environments, inequitable medical treatment, barriers to self-determination, and obstacles to economic participation [3]. Ultimately, the ADA was signed into law on July 26, 1990 (42 U.S.C. § 12101) [4]. However, despite the legislative safeguards put in place by the ADA and related laws, people with disabilities in the United States continue to face various systemic barriers that impede their access to healthcare, education, employment, and community involvement [5,6].

One of the legal rights afforded to individuals with disabilities by the ADA is the right to be accompanied by a trained service animal, defined as a dog or miniature horse that is "individually trained to do work or perform tasks" directly related to the person's disability and for the benefit of the individual (28 CFR § 36.104) [7]. In legal terms, service animals are akin to assistive technology like wheelchairs, prostheses, or hearing aids [8] and can be acquired through a professional organization or "owner-trained." The legal right of a person with a disability to be accompanied by their service animal applies in most public spaces, regardless of any existing companion animal restrictions and provided that the animal is under the handler's control. Under the ADA, employees of businesses may ask people with service animals (i.e., service animal partners) two questions to determine whether a dog or miniature horse is a service animal [9]: (1) *"Is this a service animal required because of a disability?"* and (2) *"What work or task has the dog been trained to perform?"* Service animals are not required to wear a vest or identification or to be "certified" as an indicator of legitimacy [9]. However, public access rights for service animal partners do come with contingencies. For example, service animals may lawfully be denied entry in spaces where their presence would fundamentally alter the business or entity, such as sterile environments in a hospital or certain parts of a zoo wherein seeing or smelling a service animal could disrupt the resident animals (28 C.F.R. § 36.302) [10]. However, even in these contexts, the person with a disability must still be allowed to stay and use the facility without the dog present. It is also important to note that service animal regulations may not apply to certain entities, such as certain federal or religious organizations exempt from adhering to the ADA [11].

Although legislative recognition of service animals in the United States is relatively recent, their training can be traced back to late 18th-century France [12]. Nowadays, there are tens of thousands of human-service dog dyads ("teams") reported through accrediting entities nationally and globally [13,14], and an additional unknown number of human-service dog teams trained outside of accredited schools. One factor contributing to the appeal of service dogs is their versatility; they can be trained in a wide array of tasks to aid their human partners. For example, they may guide a person who is partially sighted or blind to help them navigate independently and safely; alert a D/deaf or hard of hearing handler to sounds in their environment; assist a person with a physical disability by picking up dropped items or pulling a wheelchair; alert or respond to a medical event such as blood sugar changes for someone with diabetes; support someone with a psychiatric health condition by providing calming deep pressure input or interrupting a panic attack; and so forth [15].

Given their crucial role in their partners' lives and daily living tasks, it is unsurprising that previous research suggests that service dogs can influence their partners' social well-being [e.g., 16,17]. While service dogs are widely praised for their positive impact, some service dog partners have reported facing challenges to their social well-being due to their service dog partnership [e.g., 18,19]. For instance, in the United States, service dog partners report struggles with unwanted attention in public, stigmatization, and access denials that infringe upon their civil liberties [e.g., 20,21]. Additional reported issues include advocacy fatigue (burnout), experiencing microaggressions (subtle, indirect, or unintentional discrimination), and "handler hyper-invisibility" (wherein the service dog draws attention to the handler's disability, making them less recognized as an individual) [22]. Service dog partners with lived experiences of these challenges have encouraged research on this topic [22].

Legal inconsistencies and confusion may contribute to the concerns faced by service animal partners in the United States. These discrepancies partly arise from the fact that several federal laws address the subject of service animals, each with its own definitions and contexts. These laws include the ADA (42 U.S.C. § 12101), the Air Carrier Access Act of 1986 (ACAA; 49 U.S.C. § 41705) [23], and the Fair Housing Act of 1988 (FHA; 42 U.S.C. §§ 3601–3619) [24]. While the ACAA underwent a rulemaking process to better align its definition of a service animal with that of the ADA, crucial differences remain, leading to contradictions and conflicts that may increase barriers for people with disabilities. For example, the ACAA allows airlines to require documentation from service dog partners that must be provided at least 48 hours before travel, in contrast to the ADA, which requires no documentation except in some employment contexts (14 CFR § 382.27 2021). Meanwhile, dogs that solely provide their handlers with emotional support or offer comfort or companionship are considered companion animals (not service animals) under both the ADA and ACAA. In contrast, both service and emotional support animals fall under the FHA's broader definition of "assistance animals." These variations contribute to a situation in which an individual could easily face different rules throughout their day, placing a significant burden on service dog partners to be well-informed and adaptable to the applicable laws in each given context. Beyond the inconsistencies at the federal level, many states and cities also have their own laws regarding service animals, which can be contradictory and incompatible with federal law [25,26]. Finally, service animal-related terminology can often vary (Table 1). As this review focuses on the experiences of service animal partners in the United States, we employ the ADA terminology and definition of "service animals" and "service dogs." Ultimately, the resulting "hodgepodge of regulations" and inconsistencies in terminology and definitions have created significant confusion surrounding the topic of service animals in the United States, to the detriment of service animal partners, businesses, and the public [27].

Research on the social experiences of service dog partners remains limited; to our knowledge, no comprehensive review has been published on this topic. Therefore, this integrative review aims to synthesize empirical, theoretical, and legal literature on this subject. Our research question is: what are the social experiences reported by service dog partners in the United States, including (1) civil rights experiences, (2) experiences of stigma and discrimination, and (3) broader social experiences (e.g., social integration, participation, or support)?

## Methods

### Protocol and eligibility criteria

This preregistered integrative literature review (Open Science Framework, https://doi.org/10.17605/OSF.IO/HJ9NB) was conducted following the guidelines outlined by Toronto & Remington [29] and Preferred Reporting Items for Systematic Reviews and Meta-Analyses

**Table 1. Terminology.**

| Umbrella Terms | Commonly Understood Definition |
|---|---|
| Service animal<br>Service dog | A dog (or miniature horse) trained to do work or perform tasks for a person with a disability.<br>*Most common umbrella term in the US, as defined by the ADA.* |
| Assistance animal<br>Assistance dog | Internationally, this term is equivalent to service animal or service dog and is the most common umbrella term [28].<br>In the US, in certain situations such as housing, may also include emotional support animals (not trained in specific work or tasks). |
| **Role-Specific Terms** | **Commonly Understood Definition** |
| Guide dog | Service/assistance dog trained to help a person who is blind navigate their environment. |
| Hearing dog | Service/assistance dog trained to help a person who is d/Deaf or hard of hearing by alerting them to important sounds. |
| Mobility dog | Service/assistance dog trained to help somebody with a mobility-related disability, for example, a person who uses a wheelchair. |
| Medical alert dog | Service/assistance dog trained to alert or respond to the onset of a medical event (such as a seizure or high/low blood sugar). |
| Psychiatric service dog | Service/assistance dog trained to help a person with a psychiatric disability or neurodevelopmental disorder. |

(PRISMA) standards [30]. Institutional Review Board approval was not required as the research did not involve human subjects.

Inclusion criteria were (1) empirical or theoretical peer-reviewed publication, dissertation, thesis, or law review article; (2) if law review article, published by an American Bar Association (ABA)-accredited law school and the author has a JD, PhD, or JSD; (3) population comprises individuals with disabilities (self-reported or diagnosed) in the United States; (4) intervention defined as partnership with a service dog or service miniature horse trained to do work or perform tasks for a partner with a disability; (5) outcomes include measures of social experiences; (6) written in English. Due to recent significant revisions or clarifications of relevant federal laws, we also wanted to ensure the legal articles were relevant to the current state of service animal-related federal laws in the United States. Therefore, law review articles were further excluded if published before March 16, 2011 (if focused on the ADA) or January 12, 2021 (if focused on the ACAA), when the revisions or clarifications to the respective laws were made public.

## Search procedure

We conducted a comprehensive search across seven databases selected for their relevance to this topic: ProQuest Dissertations & Theses, ProQuest Research Library, PsycINFO, PubMed, Scopus, Nexis Uni, and Social Science Research Network. Search terms were adapted for each database (Appendix 1 in S2 File). Search results had to include one or more terms from both group 1 and group 2. Group 1 terms were service animal(s), service dog(s), assistance animal(s), or assistance dog(s). Group 2 terms were civil right(s), access denial(s), inclusion, exclusion, stigma, discrimination, attitude(s), prejudice(s), social integration, social support, social participation, social connection, loneliness, or social isolation.

We included unpublished dissertations and theses to mitigate potential bias from the "file drawer" effect, whereby studies with null or negative results are less likely to be published [31]. We further performed citation searching (i.e., a manual search of the reference lists of included articles), identifying additional relevant articles with 2,926 citations searched. This approach enabled a more complete and unbiased assessment of the available literature on social experiences reported by service animal partners.

## Screening

All articles identified through the database search were imported into Covidence software (*Covidence - Better Systematic Review Management*, 2021). This system automatically identified and eliminated duplicate items. Authors SL and MH then carried out the screening process. Items were first assessed based on their title and abstract and then based on a thorough review of the full text (proportion agreement = 0.91, Cohen's $\kappa$ = .80). Any conflicts were discussed during a weekly review meeting, with final decisions made by SL.

## Data extraction

Data were extracted from articles into a matrix according to the article type. Authors SL and MH extracted data from a randomly selected 20% of the articles to ensure sufficient agreement (proportion agreement = 0.91), after which SL extracted the remaining 80% of data. Data extracted from all articles, regardless of type, included (as applicable): authors, year of publication, title, publication type (peer-reviewed article, unpublished dissertation or thesis, or law review article), journal name, primary affiliation, article type (empirical, theoretical, or law review), objective(s), and social experience findings. From theoretical articles, we further extracted: theory/framework/model, supporting evidence, and major limitations. From empirical articles, we extracted specific aims and hypotheses, sample size, randomization, design, study time points, comparison condition, outcome measures, ethical approvals, human population, human demographics (age, gender identity, racial and ethnic identity), service animal population, service animal organization information (name, non-profit status, accreditation status, human and canine training procedures), service animal demographics (ancestry or known breed, age, origin), effect size, and major limitations. Finally, from law review articles, we extracted laws reviewed/discussed, social experience findings, and policy recommendations. Missing data were marked as missing or not reported.

## Evaluation and analysis

Article methodological rigor was evaluated following previously established practices in human-animal interaction literature reviews [32–34]. Specifically, each empirical, theoretical, and law review article was scored according to a set of relevant quality criteria, receiving 1 point for each criterion met. This resulted in a percentage score, where a higher percentage indicated higher methodological rigor. The complete list of quality criteria and references is presented in Appendix 2 in S2 File. Statistical associations between methodological rigor and publication year or article type were examined using Pearson correlations and linear regressions, respectively. Analysis of article type included publication year as a covariate. All statistical analyses were performed using R Statistical Software [35].

We completed analyses in two stages. In the first stage, we explored the magnitude of social experiences reported by service animal partners by investigating quantitative and qualitative findings reported in the empirical literature. For each of our focal areas (civil rights, stigma and discrimination, and broader social experiences), we considered (1) how many empirical studies reported findings relevant to that area, (2) how the empirical studies measured these areas, and (3) what they found. We also conducted Pearson correlations and one-way ANOVAs, respectively, to examine whether there was a relationship between publication year and reports of civil rights issues or experiences of stigma and discrimination. In the second stage, we integrated findings from empirical articles within the context and discussion presented by the theoretical and law review articles. To do so, we conducted a narrative synthesis [29,36,37]. Using our data extraction matrix, we identified common patterns across articles and examined the relationships among the data. We then iteratively organized and grouped these

themes and patterns in relation to our original research objective to identify coherent high-level themes and subthemes [37].

## Results

Following a database search conducted on September 20, 2023, we identified 465 potentially relevant items. Of these, 169 duplicates were identified and removed (147 by Covidence; 22 through screening), with an additional 238 items being excluded based on screening titles and abstracts. After full-text screening, 32 more items were excluded. Additionally, 17 more relevant articles were identified from the reference lists of the included article. A final total of 43 eligible articles are included in this review. For a visual representation of the study screening process, see Fig 1.

### Overview of articles

The 43 articles that met inclusion criteria included 23 (53%) empirical peer-reviewed articles, 11 (26%) empirical unpublished dissertations, 8 (19%) law review articles, and 1 (2%) theoretical peer-reviewed article. One law review article also reported findings from an empirical study [8]; however, because this study would not have met our criteria for inclusion on its own (it was not peer-reviewed, and participants were not service animal handlers), we

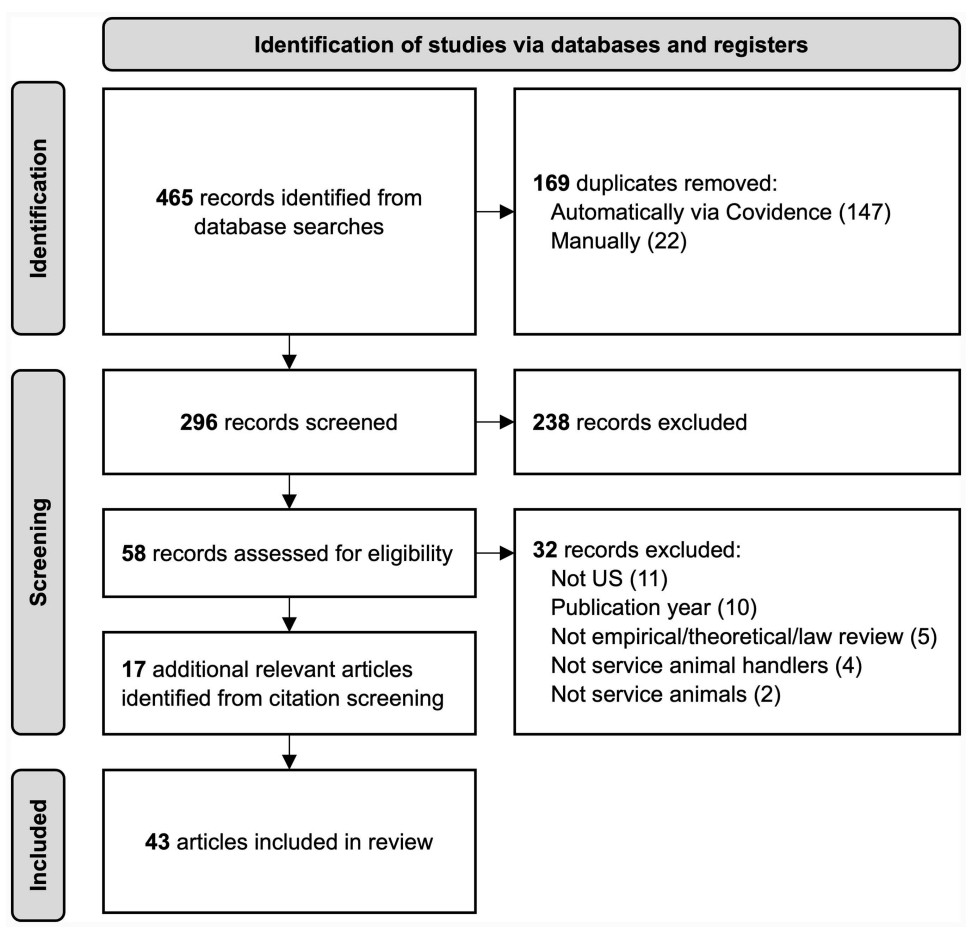

**Fig 1. Article Identification and Screening.**

considered this article exclusively as a law review for our analyses. Among empirical articles, 15 were qualitative (44%), 13 were quantitative (38%), and 6 were mixed-methods (18%). Years of publication ranged from 1987 to 2023. All articles focused solely on service dogs; none included service miniature horses. Included articles and their characteristics are summarized in Table 2.

To evaluate the data, each article received a percentage score based on how many relevant methodological rigor criteria were met. There was a wide range in terms of article scores (29% to 100%), and a significant association was found between methodological rigor score and publication year ($r = .75$, $p < .001$) in that more recently published articles had significantly higher methodological rigor (Fig 2). The top areas of methodological weakness were in reporting effect sizes (29% of quantitative empirical articles), independent assessment of participants' disabilities (36% of empirical articles), providing estimates of variability in the data for outcomes (41% of quantitative empirical articles), and describing characteristics of service animals in the study including provider and training (42% of empirical articles). There were no significant differences in methodological rigor scores between peer-reviewed publications, dissertations, or law reviews.

## Magnitude of social experiences: Empirical findings

**Civil rights experiences.** In alignment with the ADA, we defined civil rights experiences as experiences affecting the equal rights of service dog partners to access the same spaces, entities, and services as non-disabled individuals. Civil rights experiences were discussed in 35% of empirical articles [20,21,38–47] (Table 3). Of 12 eligible articles (published between 1993 and 2023), none used a standardized or validated measure to evaluate civil rights experiences. Instead, they relied on unstandardized survey questions or qualitative data collection.

In 100% of cases, reported civil rights experiences linked to service dog partnerships were self-reported access denials or service refusals; no articles noted improvements or benefits in this domain. Most of these articles (83%; 10/12) specified the percentage of participants who encountered civil rights issues, ranging from 11% to 100% of study participants. Across the cumulative sample of these ten studies, we calculated that 37% of the total participants (102/276) reported experiencing access denials or service refusals. The wide range in the number of participants reporting access denials may be partially attributable to the varied methodology employed in each study: while some studies explicitly inquired about access denials, others only recorded instances where participants brought up access issues spontaneously. Access denials were reported by participants in a broad range of settings, including business entities, air and ground transportation, schools, and workplaces [e.g., 38,40,42,43,46].

A Pearson correlation identified no significant relationship between publication year and the percentage of participants reporting civil rights issues ($r = 0.30$, [95% CI: –0.41, 0.78], $p = .399$). However, the wide confidence interval (likely due to the small sample of 10 articles) reflects high uncertainty in the estimate.

**Perceived stigma & discrimination.** We defined stigma and discrimination as negative attitudes [48] or treatment [49] of service dog partners due to their disability or – by extension – due to their status as a service dog partner. Slightly more than half of empirical articles (56%; published between 1993 and 2023) measured or discussed experiences of perceived stigma and discrimination other than (or in addition to) access or service denials (Table 3). One article used a modified version of a standardized measure, the *Everyday Discrimination Scale*, to measure self-reported discrimination [19]. No other studies employ standardized, validated measures to assess experiences of perceived stigma or discrimination. Instead, these reports were captured through open-ended surveys or qualitative surveys and interviews.

**Table 2. Summary of included articles.**

| Year | Empirical peer-reviewed article | N | Participant gender | | Service dogs |
|---|---|---|---|---|---|
| 1987 | Hart, Hart & Bergin | 28 | – | – | Mobility |
| 1988 | Eddy, Hart & Boltz | 20 | 50% F | 50% M | Mobility |
| 1989 | Mader, Hart & Bergin | 5 | 60% F | 40% M | Mobility |
| 1993 | Valentine, Kiddoo & LaFleur | 24 | 83% F | 17% M | Hearing, mobility |
| 1996 | Allen & Blascovich | 48 | 50% F | 50% M | Mobility |
| 1996 | Hart, Zasloff & Benfatto | 52 | 81% F | 19% M | Hearing |
| 2000 | Sanders | – | – | – | Guide |
| 2001 | Camp | 5 | 40% F | 60% M | Mobility |
| 2002 | Rintala, Sachs-Ericsson & Hart | 22 | 36% F | 64% M | Mobility |
| 2004 | Davis, Nattrass, O'Brien, Patronek & MacCollin | 17 | – | – | Mobility |
| 2006 | Collins, Fitzgerald, Sachs-Ericsson, Scherer, Cooper & Boninger | 152 | 62% F | 38% M | Mobility |
| 2014 | Crowe, Perea-Burns, Sedillo, Hendrix, Winkle & Deitz | 3 | 100% F | 0% M | Mobility |
| 2017 | Herlache-Pretzer, Winkle, Csatari, Kolanowski, Londry & Dawson | 4 | 100% F | 0% M | Mobility |
| 2017 | Mills | 482 | 84% F | 16% M | All types |
| 2018 | Crowe, Sánchez, Howard, Western & Barger | 9 | 0% F | 100% M | PTSD |
| 2018 | Krause-Parello & Morales | 21 | 33% F | 67% M | All types |
| 2018 | O'Haire & Rodriguez | 141 | 22% F | 78% M | PTSD |
| 2019 | Rodriguez, Bibbo, Verdon & O'Haire | 91 | 53% F | 47% M | Mobility, medical |
| 2020 | Rodriguez, Bibbo & O'Haire | 154 | 90% F | 10% M | Mobility, medical |
| 2021 | Nieforth, Rodriguez & O'Haire | 128 | 20% F | 80% M | PTSD |
| 2023 | Leighton, Rodriguez, Nieforth & O'Haire | 50 | 90% F | 10% M[a] | Autism |
| 2023 | Leighton, Rodriguez, Zhuang, Jensen, Miller, Sabbaghi & O'Haire | 168 | 26% F | 74% M | PTSD |
| 2023 | Mills | 25 | 80% F | 20% M | *Multiple types* |
| Year | Empirical Dissertation Article | N | Participant Gender | | Service Dogs |
| 1994 | Donovan | 52 | 50% F | 50% M | Mobility |
| 2006 | Rabschutz | 15 | 67% F | 33% M | Hearing, mobility |
| 2010 | Wohlfort | 11 | 64% F | 36% M | *Multiple types* |
| 2011 | Miller | 514 | 73% F | 27% M | Hearing |
| 2012 | Wild | 20 | 20% F | 80% M | Autism |
| 2014 | Newton | 6 | 17% F | 83% M | PTSD |
| 2017 | Brown | 15 | 20% F | 80% M | Autism |
| 2017 | Davis | 140 | 60% F | 40% M | Mobility |
| 2021 | Guidry | 35 | – | – | *Multiple types* |
| 2023 | Roberts | 20 | 80% F | 20% M | *Multiple types* |
| 2023 | Bristol | 5 | 100% F | 0% M[a] | Autism |
| Year | Theoretical Article | Theory | | | Service Dogs |
| 2001 | Eames & Eames | Assistance dog subculture | | | All types |

| Year | Law Review Article | Federal Laws Discussed | | | | | Context |
|---|---|---|---|---|---|---|---|
| | | ACAA | ADA | FHA | IDEA | Sec 504 | |
| 2012 | Waterlander | | ✓ | | ✓ | | Schools |
| 2016 | Huss | | ✓ | | ✓ | | Schools |
| 2016 | Lee | ✓ | ✓ | ✓ | | | Service dog fraud |
| 2017 | Huss | | ✓ | | | ✓ | Healthcare facilities |
| 2018 | Huss | | ✓ | | ✓ | | Schools |
| 2019 | Huss | | ✓ | | ✓ | | Schools |
| 2021 | Dorfman | ✓ | ✓ | ✓ | | | Service dog fraud |
| 2022 | Lally-Green, Harr Eagle & Green | | ✓ | | ✓ | ✓ | All entities |

*(Continued)*

**Table 2.** (Continued)

*Note.* – indicates not reported; F, Self-identified as female; M, Self-identified as male; PTSD, posttraumatic stress disorder; ADA, Americans with Disabilities Act of 1990; Sec 504, Rehabilitation Act of 1973 Section 504; IDEA, Individuals with Disabilities Act of 1997; FHA, Fair Housing Act of 1988; ACAA, Air Carrier Access Act of 1986.

[a]Gender identity was reported for parents of children partnered with service dogs, not the children themselves.

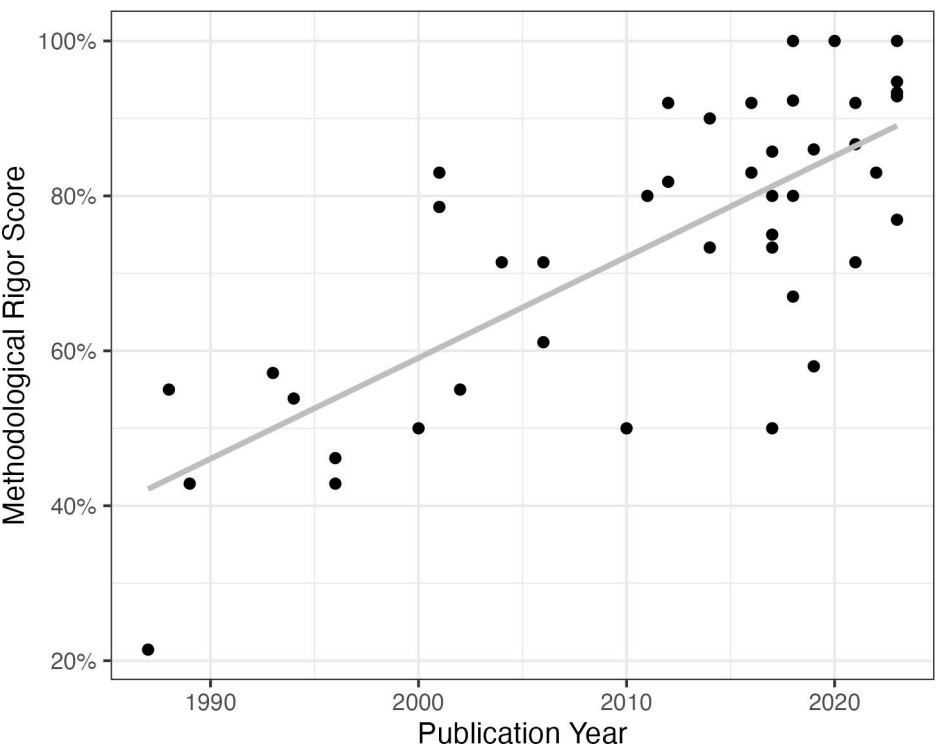

**Fig 2. Article Methodological Rigor Score and Publication Year.**

Among the 19 articles with stigma and discrimination-related findings, five studies (26%) reported service dog partners experiencing decreased stigma or discrimination compared to their experiences before service dog partnership [39,44,50–52]. After service dog partnership, these study participants felt more accepted and less judged or stigmatized by members of the public. For example, in one study, participants reported that service dog partnership shifted others' attention toward the service dog and away from their disabilities or adaptive equipment (50). Similarly, two studies of service dogs for children with autism found that participants experienced less judgment and stigma from members of the public during service dog partnerships than they had previously [51,52].

An additional four studies (21%) reported mixed findings regarding experiences of stigma and discrimination [43,45,46,53]. For example, although one study's participants described how their guide dogs could elicit stigma, they experienced guide dog partnership as less stigmatizing than using a white cane [45]. In another study, service dog partners reported barriers to public access requiring self-advocacy and education; on the other hand, they also reported a greater sense of belonging and more friendliness from strangers [46].

Finally, ten articles (53%) noted service dog partners reporting more stigma and discrimination following service dog partnership [19,20,38,41,42,54–56]. This included questioning

the service dog's legitimacy, skepticism about the individual's disability, intrusive questions, insensitive comments, staring, perceived microaggressions (e.g., telling the person they are "lucky" or that it is "cool" that they can bring their dog with them), and other customers or coworkers not wanting a dog present. Individuals seeking service dog partnerships may not always be prepared for these types of adverse experiences. For example, in one study, 44% of participants with a service dog reported adverse experiences in public, while only 22% of participants on the waitlist to receive a service dog expected to face problems [21]. Most concerningly, participants in several studies described overtly aggressive behavior from members of the public or business employees, including verbal threats and violence towards the service dog partner or their service dog [47,54]. Finally, intersectional identities may compound experiences of stigma and discrimination. For example, one study identified that younger participants and participants self-identifying as female were more likely to experience questioned legitimacy [19].

While it seems that most instances of stigma and discrimination towards service dog partnerships came from non-disabled people, this was not always the case. Stigma against service dogs can also exist within disabled communities themselves. For example, some Deaf and blind people may hold cultural stigmas against hearing or guide dog partnerships due to concerns that (1) the partnership might make them appear "disabled," whereas not all members of these communities identify as disabled in the first place, and (2) service dog partnerships could potentially generate further barriers to equal societal participation for those individuals both with and without dogs [57,58].

There was no significant association between experiences of stigma (positive, mixed, or negative) and the publication year (F(1,17) = 3.13, $p$ = .095).

**Broader social experiences.** We defined broader social experiences as any outcome related to service dog partners' social integration, participation, support, or functioning within their communities (other than experiences of access denials, stigma, and discrimination). Most empirical articles (88%; 30/34, published between 1987 and 2023) reported findings in this domain (Table 3). Out of the 16 studies with quantitative components reporting broader social experiences, 7 (44%) employed standardized, validated measures when assessing social experiences. One measure, the *Craig Handicap Assessment Reporting Technique (CHART)* Social Integration Subscale, was used in two studies (13%). All other measures were only used in one study and included: the *Community Integration Questionnaire (CIQ)*, the *Survey of Social Behavior Patterns – Friend and Social Participation Subscales*, the *Patient-Reported Outcomes Measurement Information System (PROMIS) Ability to Participate in Social Activities,* the *PROMIS Social Isolation*, the *Work Productivity and Activity Impairment (WPAI) Questionnaire*, the revised *Social Connectedness Scale (SCS-R)*, the *Pediatric Quality of Life (PedsQL)* Social Functioning and Work/School Functioning scales, and the *Adaptive Behavior Assessment System – Second Edition (ABAS-II)*.

Of these 30 articles, one (3%) reported worse social outcomes for service dog partners [56]; three (10%) reported null findings [59–61]; and 15 (50%) reported mixed findings [16,17,20,21,38,40,42–46,51,54,55,62]. Finally, 11 articles (37%) reported positive social outcomes [39,47,50,52,63–69].

Specifically, 60% of articles found that service dog partners had more social interactions or approaches, attributed to the dog's functioning as a "social bridge" promoting positive connection. However, more interaction was not always equated with a positive outcome; 33% of articles found that the extra attention due to a service dog could be inconvenient, overwhelming, or even a nuisance, requiring additional time management, planning, and anticipatory problem-solving from the service dog partner. Half of the articles assessed social interaction quality in some way, finding that interactions were generally positive, with people being nicer

**Table 3.** Empirical article social experience findings.

| Year | Peer-Reviewed Article | Service Dogs | Design | Access Issues, *n* (%) | Stigma & Discrimination | Broader Social Experiences |
|------|----------------------|--------------|--------|------------------------|------------------------|---------------------------|
| 1987 | Hart et al. | Mobility | Quantitative | – | – | Mixed |
| 1988 | Eddy et al. | Mobility | Mixed methods | – | – | Mixed |
| 1989 | Mader et al. | Mobility | Quantitative | – | – | Higher quantity/quality |
| 1993 | Valentine et al. | Hearing, mobility | Qualitative | *NS* | Mixed | Mixed |
| 1996 | Allen & Blascovich | Mobility | Quantitative | – | – | Higher participation |
| 1996 | Hart et al. | Hearing | Quantitative | 4 (11%) | – | Mixed |
| 2000 | Sanders | Guide | Qualitative | *NS* | Mixed | Mixed |
| 2001 | Camp | Mobility | Qualitative | – | Decreased | Increased participation |
| 2002 | Rintala et al. | Mobility | Mixed methods | 2 (15%) | Decreased | Mixed |
| 2004 | Davis et al. | Mobility | Qualitative | 11 (65%) | Decreased | Increased quantity/quality |
| 2006 | Collins et al. | Mobility | Quantitative | – | – | Null findings |
| 2014 | Crowe et al. | Mobility | Quantitative | – | – | Higher quantity/quality |
| 2017 | Herlache-Pretzer et al. | Mobility | Qualitative | – | Increased | Mixed |
| 2017 | Mills | All types | Quantitative | – | Increased | – |
| 2018 | Crowe et al. | PTSD | Qualitative | – | – | Increased quantity |
| 2018 | Krause-Parello & Morales | All types | Qualitative | – | Increased | Increased challenges |
| 2018 | O'Haire & Rodriguez | PTSD | Quantitative | – | – | Higher participation |
| 2019 | Rodriguez et al. | Mobility, medical | Qualitative | 10 (16%) | Increased | Mixed |
| 2020 | Rodriguez et al. | Mobility, medical | Quantitative | – | – | Higher participation |
| 2021 | Nieforth et al. | PTSD | Qualitative | 12 (17%) | Increased | Mixed |
| 2023 | Leighton … Nieforth, et al. | Autism | Qualitative | – | Decreased | Increased quantity/quality |
| 2023 | Leighton … Zhuang, et al. | PTSD | Quantitative | – | – | Mixed |
| 2023 | Mills | *Multiple types* | Qualitative | 25 (100%) | Increased | – |
| **Year** | **Dissertation or Thesis** | **Service Dogs** | **Design** | **Access Issues (%)** | **Stigma & Discrimination** | **Broader Social Experiences** |
| 1994 | Donovan | Mobility | Quantitative | – | – | Null findings |
| 2006 | Rabschutz | Hearing, mobility | Mixed methods | 11 (73%) | Mixed | Mixed |
| 2010 | Wohlfort | *Multiple types* | Mixed methods | 10 (91%) | – | Increased safety |
| 2011 | Miller | Hearing | Quantitative | – | HD stigma in D/deaf community | – |
| 2012 | Wild | Autism | Mixed methods | – | – | Higher quantity |
| 2014 | Newton | PTSD | Qualitative | 6 (100%) | Increased | Mixed |
| 2017 | Brown | Autism | Qualitative | 11 (73%) | Increased | Mixed |
| 2017 | Davis | Mobility | Quantitative | – | – | Null findings |
| 2021 | Guidry | *Multiple types* | Qualitative | – | Increased | Mixed |
| 2023 | Roberts | *Multiple types* | Qualitative | – | Mixed | – |
| 2023 | Bristol | Autism | Mixed methods | – | Decreased | Mixed |

*Note.* – indicates this article did not include outcomes in this domain. NS, percentage/number of participants reporting concerns was not specified; HD, hearing dogs.

and service dog partners receiving more smiles and friendly glances. However, again, interactions were not universally experienced as positive by service dog partners. Some articles (23%) highlighted challenges, such as others wanting to pet, feed, or distract the service dog. Additionally, in 10% of articles, some service dog partners reported feeling overshadowed by the dog, as others would greet only the dog or know the dog's name better than the human partner's name. Finally, there were mixed findings surrounding social participation. Some articles (23%) found service dog partnerships were related to more social participation, including more days of school attendance, leaving home more, and greater activity participation. In contrast, some articles reported mixed (10%), null (3%), or even negative (7%) findings, such

as problems with fraudulent service dog encounters and lower occupational self-sufficiency, more travel-related challenges, and lower odds of leaving home compared to those without service dogs.

## Narrative synthesis: integrating empirical, theoretical, and law review articles

Next, we integrated the empirical findings in the context of the theoretical and law review articles, focusing on the social challenges reported by service dog partners in the empirical literature. A narrative synthesis identified three broad themes: *Adverse Social Experiences, Contributing Factors,* and *Proposed Solutions*. Fig 3 provides a visual representation of the interconnectedness of these three themes, and Fig 4 shows a selection of exemplar quotes from participants within the reviewed articles.

**Adverse social experiences.** As discussed, while service dog partnership was associated with increased quantity and quality of the human partners' social interactions in some respects, concerns for the social well-being of service dog partners were also noted. Some service dog partners reported routinely encountering inconvenience, unwanted attention, and rude or poor behavior from others when with their service dogs in public settings. Moreover, there were numerous reports of service dog partners experiencing stigma, discrimination, and access denials in public settings – so much so that these are suggested to be defining experiences for members of an "assistance dog sub-culture" within the broader disability community [57]. This sub-culture – the service dog community – offers empathy and essential support to its members who have faced various challenges, from microaggressions and intrusive questioning to outright access denials or service refusals [54]. Beyond community support, legal cases have also arisen from access issues in various settings, with varying outcomes [e.g., 11,70].

Importantly, while service dog partners were often disadvantaged and minoritized in these situations, they were certainly not helpless victims. Service dog partners reported that

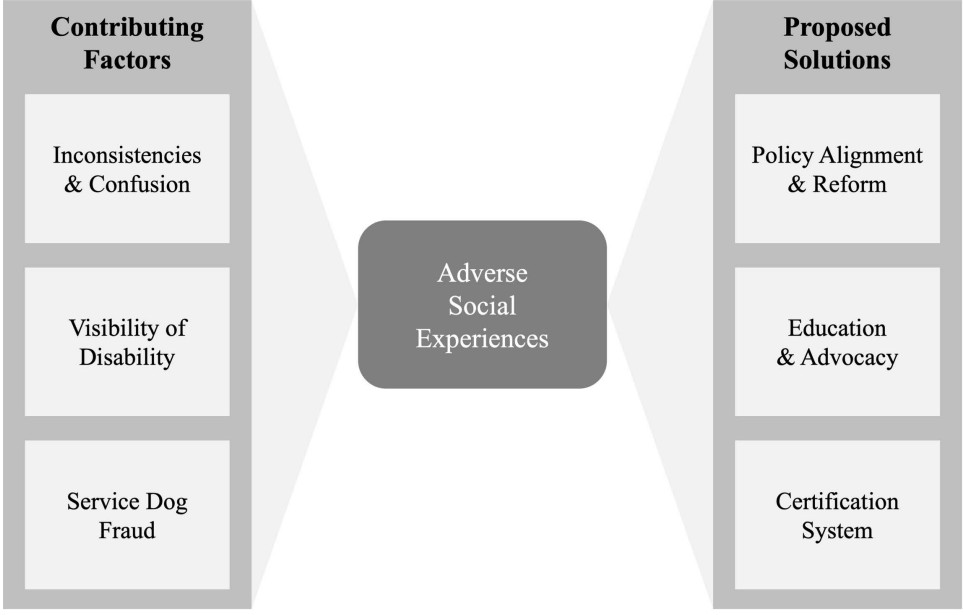

**Fig 3. Interplay of Adverse Experiences, Contributing Factors, and Proposed Solutions.**

| | |
|---|---|
| **SOCIAL INTERACTIONS** | "My service dog gives [people] something to start a conversation about – to break the ice." (39)<br><br>"A [drawback is] getting stopped by so many people. It's enjoyable at times but when I am rushed I don't want to be rude but I cut people off." (18)<br><br>"People don't always understand and are not always sensitive; you get rude questions." (37)<br><br>[I am frequently told] "you don't look disabled." (49) |
| **VISIBILITY** | "With a cane I feel horribly conspicuous and I hate using it … For some reason a dog doesn't have a stigma attached to it … With a dog I am completely me." (40)<br><br>"In the beginning I did not want a service dog because epilepsy is an invisible disease. Having a service dog now made it visible." (18)<br><br>"[Service dog] is a quick indicator for people that there is a disability present." (47) |
| **FRAUD** | "When an untrained dog goes into a public place they ruin it for people who do have a disability and do need their dog with them. You now find yourself being even more discriminated against and turned away because of others." (18)<br><br>"[Service dog fraud] is not a victimless crime … it can rob others of their independence and freedom." (36)<br><br>"When I think back on the major access disputes … in all of them, I later learned it was because of fake service dogs before me." (36) |
| **ADVOCACY & COMMUNITY** | "I would say that about 50% of the time, I'm the one that has to do the educating about everything … it's exhausting to do every single time." (48)<br><br>"I am able to [educate others about service dogs]. I don't have a problem doing it, so I see it as a gift. If I can educate as many people as I can, maybe [other handlers] won't have the problems like they do." (49)<br><br>"I'm not the only one who is having problems. It makes me feel better knowing that … and it makes it feel like it's okay. We're going to be okay." (49) |

**Fig 4. Exemplar Quotes from Service Dog Partners.**

advocating for themselves, often through educating themselves and others about their rights, enabled them to work through potential access denials in some instances [45,46,50]. Many participants also reported feeling a responsibility – for some, an intense pressure – to be good advocates and representatives for the service dog community more broadly [21,41,43]; engaging in this type of advocacy can be empowering for service dog partners and foster community advocacy [57].

However, advocacy alone may not always be sufficient to prevent access denials. For some participants, the continual experience of stereotype threat, frequent need for self-advocacy, and ongoing attention surrounding the service dog (and, by extension, the person's disability) was linked to burnout and advocacy fatigue [53,54]. In fact, service dog partners in several studies described finding negative experiences in public so unpleasant that they avoided going out in public or preferred to leave the service dog home rather than deal with these challenges, potentially compromising their health and safety [19,39,51,53]. These experiences may, at least in part, provide some explanation for the mixed social participation findings reported by empirical articles.

**Contributing factors.** *Inconsistencies and confusion.* The most-often cited factors contributing to negative experiences for service dog partners were inconsistencies and confusion surrounding laws and terminology. As previously discussed, service animals – sometimes referred to as assistance animals, guide dogs, hearing dogs, and so forth – are governed by federal and state laws, which are often misaligned. Such complexity is further compounded by numerous exceptions, conflicts, and intersections within current legislation, making it challenging for even legal experts to disentangle [71]. Even disabled handlers with legitimate, well-trained service dogs accompanying them may legally be denied access or asked to leave if the dog is not sufficiently under the partner's control. Additionally, applicable laws and procedures may vary depending on the context and entity involved – such as public and private entities, healthcare facilities, schools, and federal entities – and a person could easily be subject to multiple different sets of rules throughout a single day [e.g., 11,70,72,73]. In addition, conflicts may exist between the rights of people with different disabilities, such as those with dog allergies or phobias who require access to the same spaces as people with service dogs [8].

Relatedly, there is widespread confusion about service animals among members of the public and the "gatekeepers" responsible for determining whether a service dog's presence is legally permitted, such as business owners, business staff, and employers [e.g., 8]. Unfortunately, current laws rely on gatekeepers' assessment of the person's disability status and the service dog's legitimacy in making this determination, even though they often have no reliable way of verifying the presence of disability beyond the person's verbal assurance. This is problematic, particularly when a person's disability-related need for the dog may not be immediately apparent. Law enforcement personnel face similar challenges; while they may be called upon to address conflicts, they too lack the means to verify disability, which can contribute to disproportionate harm for those with disabilities that are not externally conspicuous (i.e., "invisible" disabilities) [74]. Moreover, as previously mentioned, several lawsuits have been brought by service dog partners following access denials. However, the courts have not always found in favor of the service dog partners in these cases, underscoring the likelihood that even service dog partners themselves are not always clear on the specifics of laws in every setting [e.g., 11,70].

Another point of discordance within service animal-related legislation relates to the roles of service animals versus emotional support animals. Emotional support animals provide companionship and emotional support to their human partners but require no specific task training. In public spaces covered by the ADA and air travel covered by the ACAA, emotional

support animals are considered equivalent to companion animals (i.e., pets), and their handlers do not have the same public access rights as disabled individuals partnered with service animals. However, in specific housing contexts covered by the FHA and workplaces covered by the ADA, emotional support animals and service animals may both be considered reasonable accommodations for persons with disabilities and subjected to contingencies on their presence [70].

*Visibility of disability.* The degree of evidence of an individual's disability may play a vital role in the social experiences they are likely to encounter when partnered with a service dog. Both research and law review articles suggest that service dog partners with invisible disabilities – disabilities that are not externally apparent, for example, diabetes or substantially disabling mental health conditions – encounter a higher frequency of discrimination relative to those with more visible disabilities [19,52–54,74]. For these individuals, partnership with a service dog may be the sole visible signal of the person's disability [40]. Unfortunately, this can lead to experiences of stigma and discrimination either because the person's disability status is questioned, because the service dog's legitimacy is questioned, because of negative attitudes towards dogs, or because of biases against individuals with disabilities [39,56]. As previously discussed, negative experiences in public can be disheartening to the point where service dog partners may avoid going out altogether or leave their dogs at home, which presents a potential health and safety risk. One research study found that this effect may be more pronounced for persons with invisible disabilities, with 53% of service dog partners with invisible disabilities reporting that they sometimes avoid going into public because of unwanted attention compared to 29% of those with visible disabilities [19].

*Service dog fraud.* Service dog fraud was commonly identified as contributing to the negative experiences encountered by service dog partners in the United States. According to one study, 84% of participants believed this was the primary reason for access denials [41]. Service dog fraud generally involves non-disabled people misrepresenting their pet dogs as service dogs to bring them into public locations where pet dogs are not typically permitted [41]. More rarely, disabled individuals may also misrepresent pet dogs as service dogs even though the dogs lack proper training and/or temperament, exhibit inappropriate behavior, or do not meet other criteria for service dog status. Despite being illegal in many states, service dog fraud remains widespread in the United States. This is due in part to (1) the restricted inquiries businesses may make of handlers, (2) limited awareness and enforcement of businesses' rights to request lawful removal of fraudulent or misbehaving service dogs, and (3) employees' desire to prevent lawsuits and avoid confrontation. Additionally, the ease with which "service dog" equipment can be purchased online and used by non-legitimate service dog teams contributes to the confusion of businesses and the public alike [8,11]. Compounding the issue, many people who engage in service dog fraud likely do not intend to harm anyone, human or animal. These individuals may simply be ignorant of the law, misconstrue rules and expectations, or overlook and discount effects on legitimate teams. Additionally, they may not perceive an identified victim of their actions, perhaps feeling instead that they are getting away with something or "sticking it to the man" [8].

Service dog fraud can have several detrimental effects. Firstly, it can devalue the hard work and education that legitimate service dog teams must undertake to work together effectively. Secondly, public safety may be risked if an untrained dog behaves aggressively, disruptively, or destructively – adversely impacting businesses, workplaces, and the public. As may happen if a legitimate service dog team is present, such misbehavior can, at best, be a distraction or annoyance. At worst, such behavior could lead to attacks on service dogs or their human partners, potentially causing injury or forcing the service dog's premature retirement [54]. Thirdly, service dog fraud can erode the public's trust in the training of service animals and

the regulations and good faith efforts that govern and shape their access to public spaces. If fraudulent service dog teams leave a negative impression on businesses, landlords, employers, or members of the public, it may result in less welcoming or even more hostile treatment towards legitimate service dog teams, leading to increased discrimination, stigma, and access denials [8,21]. Lastly and often overlooked, service dog fraud can adversely impact the welfare of companion (pet) dogs who themselves may lack the proper training, socialization, or temperament to be safe and comfortably regulated in busy public environments [52].

While the media's intensified focus on the issue of service dog fraud has raised awareness, it may also have had negative consequences for legitimate service dog teams by fueling public mistrust towards service dogs [8]. Some members of the public have taken it upon themselves to identify fraudulent service dog teams, acting as enforcers without basis or standing to assess the legitimacy of a service dog team [41]. Self-appointed enforcers may focus on "visible" indicators of legitimacy, such as the noticeability of the handler's disability, the dog's outfitting (if wearing a service dog vest), the dog's breed, or the dog's behavior, e.g., remaining inconspicuous and discreet [8,41,54]. However, there are no legal requirements in the United States that a service dog be a specific breed or that the service dog or handler be identifiable by specific equipment [8]. Additionally, while it is important for service dogs to behave appropriately when working, the expectation that a service dog should always be inobtrusive may reinforce ableist assumptions that people with disabilities should not be visible or cause inconvenience [41].

Although the problem of service dog fraud is widely recognized, federal laws currently lack a mechanism for enforcement [8,11,74]. While many states have adopted legislation to combat this issue, the feasibility of enforcing such laws remains questionable [11]. Furthermore, as previously mentioned, even if law enforcement intervenes in a suspected instance of fraud, police have no valid means of establishing the legitimacy of a service dog team [74].

**Proposed solutions.** Based on the reviewed articles, several solutions have been recommended to address concerns about the rights and well-being of service dog teams and suggested that a multi-faceted approach was likely needed [74].

*Policy alignment and reform.* Continued legislative alignment is needed to reduce confusion and discrepancies between federal and state laws [73,74]. At the state level, the Uniform Law Commission (ULC) provides a good example of promoting state law uniformity [8]. However, this alignment is needed at *both* the federal and state levels, with state laws requiring alignment not only with each other but also with federal laws. In addition to alignment, legislation must be updated to better safeguard the rights of service dog partners [38]. For instance, a more effective reporting and prosecuting system for unlawful access denials could be established [41].

Public entities, including businesses, healthcare facilities, and schools, are responsible for upholding disability-related law and safeguarding the rights of people with disabilities, including service dog partners [8]. More efforts are needed to (1) understand all parties' rights and responsibilities, including those of the entity as well as the disabled handler; (2) proactively implement policies that are clear, fair, consistent, and legally compliant; (3) provide their staff with policy implementation and enforcement training; and (4) stay educated and keep policies current as legislation changes or evolves [8,11,27,38,70,71,73]. Additionally, policies should proactively anticipate and address concerns around health and safety, service dog management, and how to accommodate individuals with conflicting needs (such as dog allergies) [73].

*Education and advocacy.* A wide range of stakeholders vitally need education and advocacy about the rights of disabled people partnered with service dogs. The general public and gatekeepers (i.e., owners or employees of entities subject to service animal-related regulations) benefit from education about service dog regulations and proper etiquette around service dog

teams; business owner and employee training, in particular, should emphasize and address potential biases [8,17,19,39,41,42,53,56,74]. Additionally, medical professionals should be educated about the impacts of service dogs to best support their clients who are interested in or have already partnered with service dogs [38,55].

There is also a need to prepare prospective service dog partners for the reality that service dog partnership will likely entail certain challenges and barriers in public settings and to provide them with education and training so that they are well-prepared to self-advocate when they encounter challenges [17,20,21,39,42]. For parents seeking service dogs for their children, there may be an additional layer of complexity involved; school administrators and educators may need targeted education and training regarding service dogs for children, and parents should ideally educate themselves on possible barriers to access in a school setting before service dog acquisition [38,71,72].

Finally, there is a need to educate the public on the consequences of service dog fraud and to draw attention to how such fraud can harm people with disabilities. Entities can help with this by using "ethical nudges" such as pop-up or text reminders when reserving travel accommodations, signage outside of shops, printed reminders on restaurant menus, etc [8].

*Certification system.*   Finally, a proposed solution to address concerns about service dog partner rights and access involves implementing a service dog certification, registration, or permitting system. This could be accomplished in a manner like the current vehicle registration system [8]. This approach could assuage the issue of entities essentially needing to assess and verify handlers' disability and disability-related need for a service dog, possibly reducing access denials and legitimacy concerns. Additionally, it could serve as a deterrent for service dog fraud [38]. Similar systems are already in place in other countries (e.g., Japan), and a small number of U.S. states (e.g., Michigan, North Carolina, and California) offer voluntary identification systems that service dog partners can opt into [8].

However, this is a complex issue with implications for civil rights, privacy, and added financial burden [8]. The proposed solution risks reinforcing the stigmatization and medicalization of individuals with disabilities, prioritizing the disabled person's responsibility to "biocertify" themselves over creating a more inclusive and accessible society [8]. Moreover, this type of system risks disproportionately burdening individuals with owner-trained (self-trained) service dogs compared to those who acquire their dog through a service dog training organization, for whom certification could be streamlined and processed by the organization on behalf of their clients [8]. Therefore, further discussion with stakeholders, particularly service dog partners, is necessary: indeed, a survey of service dog partners found that only half supported this idea [41].

## Discussion

In 1986, the National Council on Disability (formerly the National Council on the Handicapped) released a landmark report that ultimately led to the creation of the Americans with Disabilities Act. In *Toward Independence: An Assessment of Federal Laws and Programs Affecting Persons with Disabilities – With Legislative Recommendations*, the Council emphasized the need for federal law to establish "clear, consistent, and enforceable standards prohibiting discrimination" based on disability [75]. In contrast, our research indicates that the matrix of current federal laws on the rights of service dog partners in the United States is unclear, inconsistent, and difficult to enforce. As a result, some service dog partners have faced challenges in public settings, including but not limited to stigma, discrimination, and civil rights violations. This appears especially true for individuals with invisible disabilities, for whom service dog partnership effectively discloses their disability [e.g., 8,19,53,54]. This dynamic is further strained amidst the prevalence of service dog fraud, undermining the legitimacy

of trained service dog teams while eroding the public's trust in the reliability and integrity of service animals and the regulations designed to protect them and their disabled partners.

## Obstacles to inclusion and threats to civil rights

**The lived experiences of service dog partners.** The findings from this review support qualitative reports from service dog partners, who have shared their lived experiences of discrimination and marginalization [22]. In the absence of uniform construct measurement across the articles in this review, it is difficult to know the true magnitude of the aforementioned problems; however, there are clear patterns and consistent reports of adverse social experiences across a wide range of service dog placement types.

Undoubtedly, service dog partnership offers a multitude of biopsychosocial benefits that should not be disregarded: some participants in these studies reported decreased stigma, improved social interaction quantity and quality, and increased social participation, while the broader service dog literature has also identified benefits to physical and psychosocial well-being [e.g., 32,34,76]. However, service dog partnerships can also involve challenges and adverse experiences in public settings, ranging from inconvenience to overt civil rights violations, which may diminish the overall benefits. We must not simply view these challenges as inevitable consequences or "side effects" to service dog partnerships, nor rely solely on the training and preparedness of service dog partners to self-advocate. While empowering those whose well-being is at the heart of this issue is undeniably critical, the onus cannot rest solely on service dog partners – nor on service dog organizations – to solve this problem. There is no need for adverse social experiences to remain a defining experience for members of the service dog community; we can and should do better. Societally, we must take collective responsibility for facilitating access and inclusion *in the first place*.

Our research has uncovered a troubling trend of negative social experiences for service dog partners across the United States, consistent with reports from within the service dog community [e.g., 77,78]. A 2022 survey of 1,503 service dog partners worldwide found that a substantial majority (93%) had encountered suspected service dog fraud, resulting in a diminished quality of life and independence for 80% of respondents [77]. Additionally, 59% of those surveyed had experienced access denials, while 7% reported having to retire or limit their service dog's working role due to these challenges. The participants also shared that fear, anxiety, and frustration with past negative experiences had led them to limit their public outings [77]. These issues are particularly concerning and align with our findings that negative public experiences may lead to an increase, rather than a decrease, in social isolation and marginalization for some service dog partners – a stark contradiction to the spirit and goals of the ADA. Additionally, our findings appear consistent with the experiences of service dog partners in some countries outside the United States [e.g., 79,80]. Researching the social experiences of service dog partners worldwide could help identify countries that are succeeding in mitigating adverse experiences, informing legislation and policies in countries with worse outcomes.

**Health consequences.** We should also be concerned that service dog partners may be at risk for adverse health consequences as a direct result of experiencing stigma and discrimination. For one thing, some participants in the included studies reported challenges to accessing employment, housing, healthcare, and education, all of which can directly impact a person's overall well-being. Moreover, research has found that discriminatory and stigmatizing experiences can lead to compromised psychosocial well-being, social isolation, lower quality of life, and chronic stress [81,82]. Further research emphasizes the social and institutional impacts of intersectional identity [83], as individuals with multiply minoritized identities often face additional barriers, which may include nuanced negative impacts [82–84]. While one article in this review identified age and gender identity as potential

factors that could contribute to experiences of multiple discrimination, most articles did not address the effects of intersectionality or other facets of identity; this will, therefore, be an essential area for future research to explore, including capturing detailed demographics regarding the service dog partners. Additional adverse experiences could also affect disabled people whose symptoms may include disabling anxiety and distress in public spaces, such as some individuals diagnosed with posttraumatic stress disorder. For these service dog handlers, adverse experiences in public may be particularly harmful as these may exacerbate symptoms.

## Intervening to address adverse experiences

It appears evident that relying on people to "do the right thing" is insufficient, and further intervention is needed to better support service dog partners and promote social inclusion and access. Our narrative synthesis identified the need for a multi-faceted approach and collated a set of proposed solutions that intersect across socio-ecological and biopsychosocial levels. These proposed solutions can be broadly categorized into three groups: policy alignment and reform, education and advocacy, and implementation of a certification system. Achieving these goals would ideally include a single, coordinated effort rather than the current disjointed and siloed initiatives that have contributed to the present situation. A national, specialized working group – including representatives from all stakeholder groups – might be most suitable to address these issues effectively. Stakeholders include, but are not limited to, service dog partners, service dog training organizations, disability rights advocates, business entities, transportation providers, housing providers, healthcare providers, policymakers, and researchers. This group must be empowered to effect real change at all levels. It should focus on developing policies that balance safeguarding the rights of people with disabilities partnered with service dogs while minimizing burden.

**The need for policy alignment and reform.** There is a clear need for policy alignments and reforms at all levels of government - federal, state, and local. We note here that current legislation is reactionary by nature; it relies on the person with a disability to make entities aware of the need for access and accommodation rather than expecting entities to be accessible and accommodating to begin with. As identified in this review, entities and their employees are placed in the position of assessing whether the need for a service dog is valid, even while lacking the necessary expertise to make this determination. Ideally, reformed legislation should place the assessment of the need for a service dog exclusively in the hands of the disabled individual (or their caretaker, in the case of children), their healthcare provider, and (where applicable) the service dog team's organization or trainer. Further, mechanisms are needed to reliably differentiate legitimate service dogs from pets, leaving gatekeeper entities responsible solely for welcoming service dog partners and ensuring equal and equitable access. Additionally, while appropriate behavior by the service dog is certainly essential at a minimum, legislation should further emphasize the dog's working role and welfare alongside consideration of the needs of other people with and without disabilities, from those who may be inconvenienced to those who may experience adverse impacts due to disability (e.g., disabling phobias, allergies). Finally, legislation should empower public and private entities to welcome appropriately behaved service dogs in training, allowing these puppies to receive critical socialization and training in the contexts where they may work in the future [22,25]. Ultimately, once agreement is achieved at the legislative level (admittedly a monumental task in and of itself), a public awareness campaign should follow to disseminate information back to gatekeeper entities and provide training, templates, and suggestions for bringing their policies and procedures into alignment.

**Addressing and understanding service dog fraud.** Any intervention must also tackle the complex problem of service dog fraud in the United States. Based on our findings, service dog fraud is widespread and a key contributor to the adverse experiences that service dog partners face. This finding aligns with related research; for example, a 2023 survey of 77 individuals with emotional support animals in the United States found that 60% of participants had falsely presented their emotional support dog as a service dog at least once, and 18% did so frequently or almost always; this, despite the fact that emotional support dogs do not meet the defining and qualifying criteria for service dogs [85]. State legislators are actively strengthening laws around fraud; as of 2022, 33 states had enacted bans on the fraudulent representation of companion animals (i.e., pets) as service animals [86]. However, as noted, state-level efforts can lead to inconsistencies that compound problems for service dog partners rather than protect their rights and may even encourage unscrupulous businesses to take advantage of this confusion by offering fraudulent dog "certification" and equipment services [26].

While legislative efforts may be part of the solution, something currently missing from much of the conversation around service dog fraud is understanding *why* people commit service dog fraud in the first place. If fraud originates in part from the lack of an identified victim [8], shifting business signage away from "no pets allowed"-type language towards language that connects people to the "victims" who are adversely impacted may be helpful. For example, certain businesses in Arizona employ signage that states: "Please DON'T try to pass off your pet as a service animal. FAKE service animals may hurt the reputation and acceptance of valid service animals" [87]. Conversely, if service dog fraud originates partly from a desire to spend more time with one's companion animal, fraud could be disincentivized through legal and appropriate means for increased pet-owner companionship. This could be achieved through improved opportunities for pet-friendly housing, affordable and safe pet transport options, climate-controlled and secure crates for companion animals to wait for their owners outside of businesses and other similar initiatives. In summary, to more effectively address service dog fraud, it is necessary to understand underlying motives and develop responsive strategies while protecting the rights of legitimate service dog partners.

**Mechanisms for education and advocacy.** Education and advocacy are critical interventions to mitigate adverse experiences facing service dog partners. Targeted audiences should include members of the public with and without disabilities, gatekeeper entities, medical professionals, and current and prospective service dog partners. However, few articles proposed specific mechanisms for education to occur. Beyond the concerted efforts toward policy alignment and reform described above, one potential pathway for education is integrating service dog regulations and disability etiquette into state-mandated curricula. For example, nine states currently mandate that K-12 curricula include humane treatment towards animals [88]. In Oregon, this education is provided in partnership with the Oregon Humane Society; a similar model could be employed in collaboration with global service dog coalitions such as Assistance Dogs International, the International Guide Dog Federation, or the International Association of Assistance Dog Partners. For medical and mental health professionals, service dog-related education should be built directly into cultural competency training as a key component of delivering culturally congruent care to people with disabilities partnered with service dogs [89]. Additionally, service dog partners themselves should receive education and training during their partnering process to prepare them to address adverse social experiences and enable them to self-advocate. This education and training should include education surrounding relevant laws and regulations, provision of sample dialogue scripts, and role-playing of various social scenarios in a controlled setting. The service dog community will surely also continue to play a vital role as a source of support, information, and group advocacy around these challenging experiences.

**Thoughtful approaches to a certification system.** Finally, developing and implementing a service dog certification, registration, or permitting system should be carefully considered in parallel with existing or potential policy, education, and advocacy-related reforms. Although such a solution could offer many advantages, significant concerns should be anticipated and addressed. For example, a standardized verification of service dog legitimacy could drastically reduce the number of access denials, privacy impingements, and instances of fraud; however, any such system should avoid further stigmatizing or inadvertently harming service dog partners [15]. Problems with the implementation of similar systems illustrate some of these concerns; for example, voter identification laws can introduce barriers that disproportionately harm voters who are low-income, have disabilities, are racial and ethnic minorities, and the elderly [90].

In the absence of a national standard for service dog validation, some industry entities have stepped in to develop and introduce their own systems [e.g., 91,92]; however, their efficacy, impact, and scope of use are still unclear. Amidst the limited documentation requirements for service dog teams in most contexts, voluntary registry systems may confuse the public, hold little sway, and see inconsistent uptake unless and until a standardized system is implemented. Worse, they could increase public confusion, skepticism, and questioning of service dog partners who do not carry such identification.

One possible avenue to consider may be certification of service dog training processes, rather than the service dog teams themselves. Under this system, service dogs in the United States would be required to wear equipment identifying that the dog was trained through an accredited training process. If such a system were implemented in an accessible and equitable manner, it could shift some of the added burden onto organizations and private trainers rather than individuals with disabilities themselves. This approach could contribute to high training and behavioral standards for service dogs and reduce issues of access denials, questioned legitimacy, and fraud. However, nuanced challenges would need to be addressed, such as consideration for owner-trained service dogs, visiting international service dog partners, and service dogs in training. Ultimately, further discussion among stakeholders is necessary to inform any decisions regarding the development and implementation of such systems.

## Future directions

**Intentionally measuring adverse social experiences.** The methodological rigor of articles was significantly positively associated with publication year, indicating that the quality of studies has improved over time – a promising trend for the field of human-animal interaction more broadly. That said, very few of the empirical studies in this review – particularly quantitative studies – intentionally measured adverse social experiences such as stigma, discrimination, or access denials. The lack of routine measurement of these constructs makes it difficult to accurately assess the true scope of these adverse experiences. It also underscores the value of conducting community-based participatory research to identify key issues in need of investigation [e.g., 93]. Moving forward, researchers should begin operationalizing the challenges associated with service dog partnerships, routinely include measures to capture these outcomes, and include effect sizes when reporting outcomes. This will not only enable us to develop a more nuanced understanding of the magnitude of these problems, but it will also allow us to track and measure changes over time that result from implementing any solutions addressing these concerns.

One option is to incorporate standardized, validated self-report measures into research, such as the *Everyday Discrimination Scale* [94], used in one of the studies in this review. However, this scale includes both general items (i.e., perceived respect) and context-specific items (i.e., "treated worse in a restaurant or store"; "considered less smart"), whereas ideally, a scale

would only have items that can apply to all contexts. Moreover, there is a need to continually develop appropriately normed measures and bring study recruitment and assessments in line with best practices and recommendations for the inclusion of people with disabilities [95,96]. Therefore, a possible future direction includes the development and validation of a standardized scale to assess adverse social experiences faced by service animal partners. Additionally, given the call for greater use of ecological momentary assessment and similar methodologies in human-animal interaction research [e.g., 97], researchers examining social outcomes for service dog partners may consider asking questions about adverse social experiences via ecological momentary assessments, daily diaries, or as part of day reconstruction methods [98,99]. Finally, researchers can help by pilot testing and empirically investigating potential solutions; for example, it would be helpful to study whether different types of wording on storefront signage are more or less effective in mitigating service dog fraud. International and cross-cultural studies spanning nations with different legislation or education around service dogs may also help to identify strengths and limitations of particular legal or policy approaches in this area.

**Impacts to service dogs.** While not the focus of the present review, it is also crucial to recognize that adverse experiences in public affect not only human service dog partners but also the service dogs themselves. As mentioned, encounters with stressed and possibly aggressive companion animals being fraudulently misrepresented as service dogs can pose serious safety risks for legitimate service dogs. More subtly, service dogs may also perceive and be affected by their human partners' and other humans' and animals' stress [e.g., 100] and may be at greater risk for adverse health outcomes if frequently left at home without their human partners [39]. Service dogs are typically carefully screened and selected for their working roles [101]; if it is the case that they experience benefits when performing their work, it is plausible that they are also negatively impacted when prevented from performing their job. However, none of the articles in this review measured the impacts of adverse social experiences on the service dogs themselves. We, therefore, echo calls from the field of human-animal interaction [e.g., 102] for future studies to include measures of both human *and* animal well-being as part of the outcomes assessed.

**Challenging ableism.** Lastly, we emphasize that the stigma, discrimination, and access denials that service dog partners encounter do not occur in a vacuum; in many ways, these are likely manifestations of individual and institutional ableism and the systemic marginalization of people with disabilities. Ongoing work is needed to systematically identify and address the root causes and consequences of ableism. Indeed, the findings from this review are somewhat aligned with research suggesting that the use of assistive devices (to which service dogs are legally equivalent) may engender experiences of perceived stigmatization and increased barriers, potentially even facilitating their disuse altogether [103–105]. However, there is no mention in these studies of experiences of access denials, questioned legitimacy, or others trying to distract or "pet" the assistive device. This implies that service dog partnerships and service dogs differ from assistive devices in important and unique ways. This said, the researchers of these articles emphasize the importance of design "for social acceptability" [105]. In this regard, there may be reason for optimism regarding service dog partnerships: despite adverse social experiences, research has found that most people hold positive implicit and explicit attitudes and perceptions towards both service dogs themselves and people with disabilities accompanied by service dogs [106–109]. With growth in society's understanding of the relationship between people with disabilities and their service dogs, there is tremendous potential to challenge misconceptions and effect powerful change. While facilitating empathy, education, and inclusive policies, we can collectively nourish a culture that celebrates diversity and plurality, embracing the fuller spectrum of human ability.

## Limitations

Several limitations of this integrative review should be considered. Firstly, as service animal-related laws may vary across different countries, our findings may not apply to service dog partners, the public, or businesses outside the United States. Secondly, studies captured self-report information about service dog partner experiences of perceived stigma, discrimination, and access denials. There is no way to verify the accuracy of these claims; this is particularly relevant in the case of access denials, given that there are legally valid reasons for denying access under certain circumstances. Furthermore, there is no way to determine what exact form of access denial took place: the service dog team being unlawfully turned away altogether or the service dog being denied access (whether lawfully or not) while the handler was allowed to stay without the dog present. Thirdly, while we planned to include service dogs and miniature horses in this review, there were no articles about service miniature horses, possibly due to lower prevalence; findings may not be generalizable to people with disabilities partnered with service miniature horses. Finally, while we tried to capture all relevant articles by searching multiple databases, employing broad search terms, and searching reference lists of included articles, it is possible that additional relevant and eligible articles could have been missed.

## Conclusion

This integrative review found that, although service dog partnerships can be associated with positive social experiences, many service dog partners in the United States have also reportedly experienced social marginalization in the form of stigma, discrimination, and access denials. Moreover, service dog partners with invisible disabilities may be at higher risk for adverse social experiences. The lack of adequate legal protections at federal and state levels and the high prevalence of service dog fraud contribute to these issues. A multi-component solution, operating at all socio-ecological levels, is necessary to mitigate these challenges effectively. Overall, the findings from this review emphasize the need to reduce barriers and increase access and inclusion for the well-being of service dog partners.

## Supporting Information

**S1 Checklist. PRISMA Checklist.**
(PDF)

**S2 File. Appendices 1 and 2.**
(DOCX)

## Acknowledgments

Thank you to Stephanie Bristol, Wallis Brozman, Clare Jensen, Leanne Nieforth, and Kristy vanMarle for their insightful comments and feedback, and to Briana Delossantos for her support.

## Author contributions

**Conceptualization:** Sarah C. Leighton, Matthias R. Mehl.

**Data curation:** Sarah C. Leighton.

**Formal analysis:** Sarah C. Leighton.

**Funding acquisition:** Sarah C. Leighton.

**Investigation:** Sarah C. Leighton, Molly E. Hofer.

**Methodology:** Sarah C. Leighton, Molly E. Hofer, Cara A. Miller, Matthias R. Mehl, Tammi D. Walker, Evan L. MacLean, Marguerite E. O'Haire.

**Project administration:** Sarah C. Leighton.

**Resources:** Matthias R. Mehl, Marguerite E. O'Haire.

**Supervision:** Matthias R. Mehl, Tammi D. Walker, Evan L. MacLean, Marguerite E. O'Haire.

**Writing – original draft:** Sarah C. Leighton.

**Writing – review & editing:** Sarah C. Leighton, Molly E. Hofer, Cara A. Miller, Matthias R. Mehl, Tammi D. Walker, Evan L. MacLean, Marguerite E. O'Haire.

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
