## [Decision Letter · Decision Letter 0]

6 Aug 2024

PONE-D-24-23942Obstacles to inclusion and threats to civil rights: An integrative review of the social experiences of service dog partnersPLOS ONE

Dear Dr. Leighton,

Thank you for submitting your manuscript to PLOS ONE. After careful consideration, we feel that it has merit but does not fully meet PLOS ONE’s publication criteria as it currently stands. Therefore, we invite you to submit a revised version of the manuscript that addresses the points raised during the review process.

Please kindly address the reviewers' minor suggestions for additional methodological clarity.

We look forward to receiving your revised manuscript.

Kind regards,

Avanti Dey, PhD

Staff Editor

PLOS ONE

Journal Requirements:

Additional Editor Comments (if provided):

Reviewers' comments:

Reviewer's Responses to Questions

**Comments to the Author**

1. Is the manuscript technically sound, and do the data support the conclusions?

Reviewer #1: Yes

Reviewer #2: Yes

2. Has the statistical analysis been performed appropriately and rigorously? 

Reviewer #1: Yes

Reviewer #2: Yes

3. Have the authors made all data underlying the findings in their manuscript fully available?

Reviewer #1: Yes

Reviewer #2: Yes

4. Is the manuscript presented in an intelligible fashion and written in standard English?

Reviewer #1: Yes

Reviewer #2: Yes

5. Review Comments to the Author

Reviewer #1: Overall, this is an excellent manuscript and very well-written. It is quite lengthy, but I’m not sure if that needs to change because everything you’ve included is well-justified. I do wonder about your intended audience, though. I believe this article meets all of the journal publication requirements. I have provided additional points of feedback below.

Title: given the exclusive focus on American-based research, this manuscript title would benefit from reflecting this in some way.

Lines 76-79: “For example, service animals may lawfully be denied entry in spaces where their presence would fundamentally alter the business or entity, such as sterile environments in a hospital or certain parts of a zoo wherein seeing or smelling a service animal could disrupt the resident animals.” – It would be helpful to include the legislation code here or a citation.

Lines 83-85: “Although legislative recognition of service animals in the United States is relatively recent, training service animals – specifically, guide dogs – can be traced back to late 18th century France.” – Given the myriad terms, it would be helpful to explain somewhere exactly which categories fall under “service dog”, beyond just guide dogs. Also, how do “assistance dogs” relate? I see mention later on of service and emotional support animals falling under the broader definition of “assistance animals” except in other publications service and assistance animals are considered synonyms.

Lines 85-86: “Nowadays, there are tens of thousands of human-service dog dyads (“teams”) both nationally and globally (10,11)”: You have only sited ADI and IGDF who solely publish on their membership data. As such, this estimate may not be wholly accurate. I believe a comment on this should be included.

Lines 132-136: How does your article differ from others on this topic? Has anything like this been published before? How does your research uniquely contribute to this area? The word “impact” implies causality, and many of the articles you extracted do not examine causal links. Given the methods and results of your article, your research questions do not fully reflect what you examined and found. I would recommend clarifying that you questioned: “What are the ‘reported/published’ social effects faced by…..”

Methods: Is this type of research exempt from review with your institutional research ethics board?

Line 145: “(3) population comprises individuals with disabilities in the United States” - are disabilities self-determined or do they need to be medically determined?

Lines 159-160: Inclusion criteria included “service miniature horse”, so why was this not a search term?

Lines 202-204: Inclusion criteria included “service miniature horse”, so why were only the social impacts of service dog partners analyzed?

Lines 284-286: Regarding the definitions of stigma and discrimination you adopted, do these reflect the published literature and consensus on these constructs?

Lines 520-522: Can you please clarify what you mean by “…study participants of the reviewed articles recommended…” – do you mean you’re reporting on what past study participants suggested in the articles you examined? Or did you somehow contact past study participants to get their views?

Reviewer #2: This is a comprehensive, well-done review of the social experiences of service dog partners.

- To improve readability it is suggested that sub-headings be added to the Discussion section of the manuscript.

-If available, quotes from service dog partners may enhance the impact of the findings. These could either be in the literature review section or in a table.

6. PLOS authors have the option to publish the peer review history of their article (what does this mean? ). If published, this will include your full peer review and any attached files.

**Do you want your identity to be public for this peer review?** For information about this choice, including consent withdrawal, please see our Privacy Policy .

Reviewer #1: No

Reviewer #2: No

---

## [Author Response · Author response to Decision Letter 1]

6 Sep 2024

Thank you for reviewing our manuscript. Please see the attached Response to Reviewers for detailed responses to reviewer comments.

---

## [Decision Letter · Decision Letter 1]

1 Nov 2024

Obstacles to inclusion and threats to civil rights: An integrative review of the social experiences of service dog partners in the United States

PONE-D-24-23942R1

Dear Dr. Leighton,

We’re pleased to inform you that your manuscript has been judged scientifically suitable for publication and will be formally accepted for publication once it meets all outstanding technical requirements.

Kind regards,

Laura Kelly, PhD

Division Editor

PLOS ONE

Additional Editor Comments (optional):

Reviewers' comments:

Reviewer's Responses to Questions

**Comments to the Author**

1. If the authors have adequately addressed your comments raised in a previous round of review and you feel that this manuscript is now acceptable for publication, you may indicate that here to bypass the “Comments to the Author” section, enter your conflict of interest statement in the “Confidential to Editor” section, and submit your "Accept" recommendation.

Reviewer #1: All comments have been addressed

Reviewer #2: All comments have been addressed

2. Is the manuscript technically sound, and do the data support the conclusions?

Reviewer #1: Yes

Reviewer #2: Yes

3. Has the statistical analysis been performed appropriately and rigorously? 

Reviewer #1: N/A

Reviewer #2: Yes

4. Have the authors made all data underlying the findings in their manuscript fully available?

Reviewer #1: Yes

Reviewer #2: Yes

5. Is the manuscript presented in an intelligible fashion and written in standard English?

Reviewer #1: Yes

Reviewer #2: Yes

6. Review Comments to the Author

Reviewer #1: Thank you so much for your responses and addressing my questions and feedback. I have no further comments.

Reviewer #2: (No Response)

7. PLOS authors have the option to publish the peer review history of their article (what does this mean? ). If published, this will include your full peer review and any attached files.

**Do you want your identity to be public for this peer review?** For information about this choice, including consent withdrawal, please see our Privacy Policy .

Reviewer #1: No

Reviewer #2: No

---

## [Editor Report · Acceptance letter]

PONE-D-24-23942R1

PLOS ONE

Dear Dr. Leighton,

I'm pleased to inform you that your manuscript has been deemed suitable for publication in PLOS ONE. Congratulations! Your manuscript is now being handed over to our production team.

Kind regards,

on behalf of

Dr. Laura Hannah Kelly

Staff Editor

PLOS ONE